# A General Framework for Synthesizing and Executing Self-Explaining Plans for Human-AI Interaction

**Sarath Sreedharan[1], Tathagata Chakraborti[2], Christian Muise[2] , Subbarao Kambhampati[1]**
[1]CIDSE, Arizona State University, Tempe, AZ 85281 USA
[2]IBM Research AI, Cambridge, MA, USA
ssreedh3@asu.edu, tathagata.chakraborti1@ibm.com, christian.muise@ibm.com, rao@asu.edu

## Abstract

In this work, we present a general formulation for decision making in human-in-the-loop planning problems where the human's expectations about an autonomous agent may differ from the agent's own model. We show how our formulation for such multi-model planning problems allows us to capture existing approaches to this problem and also be used to generate novel explanatory behaviors. Our formulation also reveals a deep connection between multi-model planning and epistemic planning and we show how we can leverage classical planning compilations designed for epistemic planning for solving multi-model planning problems. We empirically show how this new compilation provides a computational advantage over previous approaches that separate reasoning about model reconciliation and identifying the agent's plan.

## 1 Introduction

As automated agents and users start working closely together, it becomes increasingly important that the agents are capable of acting in a manner that is intuitive and explicable to users in the loop. A major challenge to achieving such fluent collaboration is the fact that the human's expectations regarding the agent's capabilities and preferences may differ from reality. Such knowledge asymmetry implies that even in cases where the human teammate is a passive observer, the agent can no longer solely reason with their individual models to generate desirable plans. Instead, the agent needs to explicitly take into account the user's expectations about the agent when coming up with its plans. Previous works have mostly focused on two strategies to handle such scenarios, (a) explanation - the agent chooses to perform its optimal plan and explain the effectiveness of the chosen plan; (c.f [Chakraborti et al., 2017]) (b) explicable planning - the agent chooses to follow viable plans that are closest to user's expectations (c.f [Zhang et al., 2017]).

In the end, we would want an approach that is able to combine the strengths of these two strategies. This would require the agent to move away from standard notions of decision making where the agent is solely trying to optimize the cost of the plan it will follow, but also take into account the ease of explaining the plan as one of the criteria for choosing its actions. We will refer to the problem of generating such plans as **multi-model planning**.

In this work, we will present a general characterization of the problem of multi-model planning and discuss how we could view the solutions to such multi-model planning problems as **self-explaining plans**, where explanations are themselves provided through robot actions. These actions could be purely communicative actions that are meant to update the human's mental model or task level actions that could also have epistemic side effects. The contributions of this paper include,

- Presenting a formalization of multi-model planning that allows us to characterize solutions identified by earlier works (Section 2).

- We look at two additional problem considerations; the phase of interaction (is the explanation occurring during plan selection or is it happening during plan execution); the attentiveness of the user (Section 5) and discuss how such considerations poses both new challenges and provides us with opportunities to generate novel behaviors.

- We present a new planning compilation to solve such planning problems that allows for a uniform treatment of explanation and task level actions. We empirically show how such a compilation could provide a computational advantage (Sections 6 and 7).

We will use Urban Search and Rescue domain as a running example to motivate and illustrate the discussions.

## 2 Multi-Model Planning and Explanation as Model Reconciliation

The planning models used by both the human and the robot are described by the tuple $\mathcal{M} = \langle F, A, I, G \rangle$. In this formulation, $F$ is the set of propositional fluents used to describe the planning task states, $A$ the set of actions, $I$ the initial state and $G$ the goal. Each action $a \in A$ is further defined as a tuple of the form $a = \langle \text{prec}^a, \text{adds}^a, \text{dels}^a \rangle$, where $\text{prec}^a$ lists the preconditions of the action and $\text{adds}^a$ and $\text{dels}^a$ provides the add and delete effects of the action. In general, the precondition can be some logical formula defined over state fluents and an action $a$ can only be executed in a state $S$ if $S \models \text{prec}^a$. The effects are generally of the form $c \to e$, where the antecedent

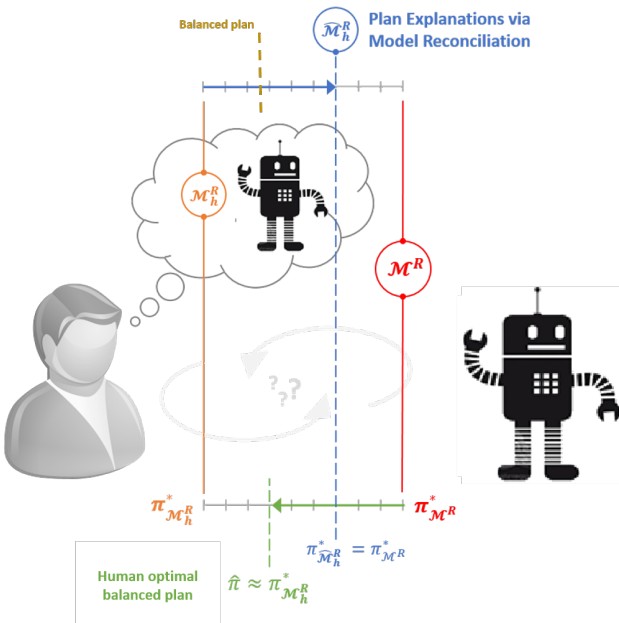

Figure 1: An overview of planning in multi-model planning settings.

represents the condition under which the effect $e$ should be applied (where the fluent corresponding to $e$ is set true in the state if $c \rightarrow e$ is part of the add effects and if it is part of the delete it is set to false).

Each action is also associated with a cost (represented as $C_{\mathcal{M}}(a)$). A plan in this setting is defined as a sequence of actions ($\pi = \langle a_1, ..., a_n \rangle$) and is said to be valid (denoted as $\pi(I) \models_{\mathcal{M}} G$) for a model $\mathcal{M}$ if $G \subseteq \pi(I)$. Each plan is additionally associated with a cost $C_{\mathcal{M}}(\pi)$ such that $C_{\mathcal{M}}(\pi) = \sum_{i=1}^{n} C_{\mathcal{M}}(a_i)$. A plan $\pi$ is said to be optimal if there exist no valid plan $\pi'$ such that $C_{\mathcal{M}}(\pi') < C_{\mathcal{M}}(\pi)$. We will use $\Pi^*_{\mathcal{M}}$ to represent the set of all plans optimal for $\mathcal{M}$.

In this setting we will assume that the robot uses a model $\mathcal{M}_R = \langle F, A_R, I_R, G_R \rangle$ and the human evaluates the plan using a model $\mathcal{M}_H = \langle F, A_H, I_H, G_H \rangle$. For ease of discussion, we concentrate on the specific case where conditions for actions only consist of conjunction of positive literals and with no action cost difference between models.

We start with the assumption that the robot is aware of $\mathcal{M}_H$ and hence knows whether a given plan $\pi_R$ (that is optimal in $\mathcal{M}_R$) is **explicable** or not, i.e, whether or not the human would identify $\pi_R$ to be one of the optimal plans for the given planning problem. In cases where the given plan may appear inexplicable, one way the robot could resolve the confusion would be by informing the human about its own model so they can correctly evaluate the current plan. Thus an explanation ($\mathcal{E}$) for this setting can be represented by a set of model updates (where $\mathbb{E}$ represents the set of all possible model updates). The different types of model updates include –

(1) Turn a fluent p true or false in initial state (represented by the operator {add/remove}-p-from-I)
(2) Add or remove a fluent p from the precondition (also add or delete) list of an action a (represented by the opera-

tor {add/remove}-p-from-prec-of-a)
(3) Add or remove a fluent p from the goal list (represented by the operator {add/remove}-p-from-G)

We will use the function $\mathcal{T} : \mathbb{M} \times \mathbb{E} \rightarrow \mathbb{M}$ to represent the transition function induced by the model update messages (where $\mathbb{M}$ represents the set of all possible models). A set of model updates ($\mathcal{E}$) **explains** a given plan if in the model resulting from applying the model updates ($\hat{\mathcal{M}} = \mathcal{T}(\mathcal{M}_H, \mathcal{E})$), the current plan is optimal (i.e., $\pi_R \in \Pi^*_{\hat{\mathcal{M}}}$). Unless otherwise mentioned, when we refer to **updated model**, we are referring to this new human model obtained by applying the explanations. Once we have such a set of model updates, the final explanation (presented to the explainee) can be generated by converting the model updates to corresponding natural language statements [Tellex et al., 2014] or through some appropriate visualization [Chakraborti, Sreedharan, and Kambhampati, 2018a].

Among the valid explanations for a given plan, we refer to the shortest explanation as the minimally complete explanation or MCE. The original work [Chakraborti et al., 2017] on model reconciliation viewed the problem of generating MCE explanation as a problem of searching over the space of possible model updates that can be performed on the human model and the validity of each possible explanation was measured by checking the optimality of the plan in the corresponding planning problem. Before we go into more details, let us briefly look at the urban search and rescue (USAR) domain that will act as the running example for the rest of the paper.

## 3 Urban Search and Rescue

USAR presents an ideal testbed for research on explainable planning as it looks at cases where the decision to follow suboptimal or in-executable plan can be potentially disastrous, yet limitations in communications capability could prevent the agents from providing detailed explanations.

The basic scenario consists of an autonomous agent that has been deployed to the disaster scene and an external commander who is monitoring the activities of the robot. Both agents start with the same model of the world (i.e the map of the building before the disaster) but the models diverge over time owing to the fact that robot has access to more accurate information about the current status of the building. This model divergence could lead to the commander incorrectly evaluating valid robot plans as sub-optimal or unsafe. One way to satisfy the commander would be to point out possible changes to it's model that led the robot to come up with the plan in the first place.

Figure 2 illustrates a typical scenario where the robot needs to travel from P1 to its goal at P15. Here the human believes the robot should be moving to waypoint P6 and follow that corridor to go to P15, while the robot knows it should be moving to P7. This disagreement rises from the fact that the human incorrectly believes that the path from P6 to P5 is clear while that from P8 to P12 is blocked. If the robot were to follow the explanation scheme that was established in [Chakraborti et al., 2017] then the robot would stick to its own plan and provide the following explanation –

```
> remove-(clear p6 p5)-from-I
```

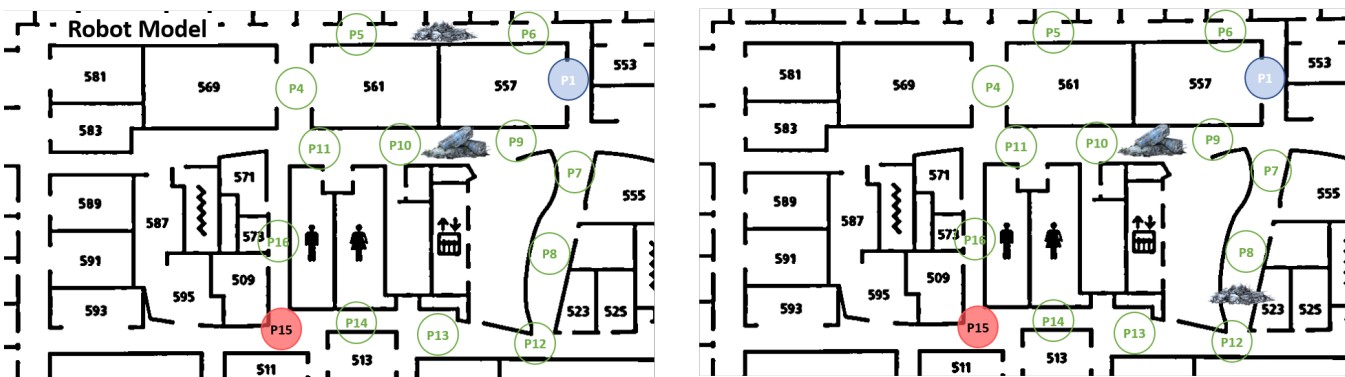

Figure 2: The basic robot and human maps. The robot starts at P1 and needs to go to P15. The human incorrectly believes that the path from P6 to P5 is clear and the one from P8 to P12 is blocked. Both agents know that there are some movable rubble between P9 and P10 that can be moved with the help of a costly clear_passage action.

```
      (i.e., Path from P6 to P5 is blocked)
> add-(clear p8 p12)-to-I
      (i.e., Path from P8 to P12 is clear)
```

## 4 Multi-Model Planning

In this work, we will be looking at the more general case where we are interested in identifying both the agent plan and the relevant explanation, and we will refer to this problems as multi-model planning

**Definition 1.** *A **multi-model planning problem** is defined by the tuple* $\Psi = \langle \mathcal{M}_R, \mathcal{M}_H, \mathcal{T} \rangle$, *where* $\mathcal{M}_H$ *is the human model,* $\mathcal{M}_R$ *the robot model and* $\mathcal{T}$ *is the transition function. A solution to the problem* $\Psi$ *is given by the tuple* $\langle \mathcal{E}^\Psi, \pi_\Psi \rangle$, *where* $\mathcal{E}^\Psi$ *is set of model updates and* $\pi_\Psi$ *a plan, such that* $\pi_\Psi(I_{\mathcal{M}_R}) \models_{\mathcal{M}_R} G_{\mathcal{M}_R}$ *and* $\pi_\Psi(I_{(\mathcal{T}(\mathcal{M}_H, \mathcal{E}^\Psi))} \models_{\mathcal{T}(\mathcal{M}_H, \mathcal{E}^\Psi)}$ $G_{\mathcal{T}(\mathcal{M}_H, \mathcal{E}^\Psi)}$.

The above problem only deals with cases where we care about establishing the validity of a robot plan in the human model. The above problem is already **PSPACE-complete** [1], but we need to go beyond just finding valid solutions to finding explanations that establish the optimality of a robot plan in the human model.

**Definition 2.** *The tuple* $\langle \mathcal{E}^\Psi, \pi^\Psi \rangle$ *is said to be a complete solution for the problem* $\Psi$ *if* $\pi^\Psi \in \Pi^*_{\mathcal{M}_R}$ *and* $\pi^\Psi \in \Pi^*_{\mathcal{T}(\mathcal{M}_H, \mathcal{E}^\Psi)}$.

The use of the term **complete** here is in accordance with it's use in [Chakraborti et al., 2017]. $\mathcal{E}^\Psi$ constitutes an MCE for a plan $\pi_\Psi$ when there exists no solution of the form $\langle \mathcal{E}^\Psi_2, \pi_\Psi \rangle$, such that $\mathcal{C}_\mathcal{E}(\mathcal{E}^\Psi_2) < \mathcal{C}_\mathcal{E}(\mathcal{E}^\Psi)$, where $\mathcal{C}_\mathcal{E}$ is the cost of providing an explanation. The use of complete solutions are best suited for cases, where the agents choose to stick to a plan that is optimal and explain away any confusion an observer may have regarding the chosen plan. While the agent could choose to follow the optimal plan that is easiest to explain,

in many scenario, communicating the explanation could still constitute a considerable expense on the agent's part. It may in fact be more desirable for the agent to follow a sub-optimal plan if the choice of plan results in lower cost for transmitting the explanations. This is a trade-off we make frequently in our day-to-day life and it would be desirable for an explicable agent to be capable of balancing these two sources of cost (i.e the cost of the plan being followed and the cost of communicating the corresponding explanations) [Chakraborti, Sreedharan, and Kambhampati, 2018b].

**Definition 3.** *For a problem* $\Psi$, *the tuple* $\langle \mathcal{E}^\Psi, \pi^\Psi \rangle$ *is said to be the optimal balanced explicable solution if,*

1. $\pi^\Psi(I_R) \models G_R$.

2. $\pi^\Psi \in \Pi^*_{\mathcal{T}(\mathcal{M}_H, \mathcal{E}^\Psi)}$.

3. $\nexists \langle \hat{\mathcal{E}}, \hat{\pi} \rangle$, *such that the tuple satisfies (1.) and (2.), and* $C(\hat{\mathcal{E}}) + C_{\mathcal{M}_R}(\hat{\pi}) < C(\mathcal{E}^\Psi) + C_{\mathcal{M}_R}(\pi^\Psi)$.

In the USAR domain, to choose a balanced plan the robot can follow the path through P9 that also involves clearing the movable rubble between P9 and P10. In this case the robot only needs to provide a single explanation.

```
> remove-(clear p6 p5)-from-I
      (i.e., Path from P6 to P5 is blocked)
```

The original work that studied the problem of balancing (c.f [Chakraborti, Sreedharan, and Kambhampati, 2018b]) used an additional hyperparameter $\alpha$, that captures the agent's relative preference to providing explanations versus following a costlier explicable plan. In our formulation, we assume that such preferences are automatically captured by the cost of the explanation. This formulation allows us to better capture the fact that certain aspects of the model may be harder to communicate than others. Conceptually, this formulation should also allow us to include the effort required at the user's end into the explanation cost, but capturing such considerations faithfully would require us to go beyond simple additive models cost explanation cost and so we will ignore them in the current work.

Note that the above formulation looks for solutions where the given plan is completely explicable to the user. In fact

---

[1]We can compile plan existence problems into a model reconciliation problem and as shown in section 6 we can compile the problem of finding a solution to plan existence in a conditional planning problem which is again PSPACE hard [Nebel, 2000]

the above formulation doesn't allow for the generation of purely explicable plans of the type discussed in [Kulkarni et al., 2019; Zhang et al., 2017], which tries to identify plans closest to the human optimal plan without relying on any explanations. In many cases, generating such plans that may just be "good-enough" may be more desirable to the agent than always trying to stick to completely explainable plans (at the cost of providing more explanation). We can do this by ensuring that the agent could include the extend of inexplicability of a given plan as yet another metric to consider when generating its plans rather than holding it as hard constraint. Assuming that the degree of inexplicability of the plan is directly proportional the degree of suboptimality of the plan in the human model, we can define an optimal balanced solution as follows.

**Definition 4.** *For a problem $\Psi$, the tuple $\langle \mathcal{E}^\Psi, \pi^\Psi \rangle$ is said to be the optimal balanced solution if,*

1. $\pi^\Psi(I_R) \models G_R$.

2. $\nexists \langle \hat{\mathcal{E}}, \hat{\pi} \rangle$, *such that the tuple satisfy (1.) and*
   $(C(\hat{\mathcal{E}}) + C_{\mathcal{M}_R}(\hat{\pi}) + \beta * (C_{\mathcal{T}(\mathcal{M}_H, \hat{\mathcal{E}})}(\hat{\pi}) - C_{\mathcal{T}(\mathcal{M}_H, \hat{\mathcal{E}})}(\pi_1^*))) < (C(\mathcal{E}^\Psi) + C_{\mathcal{M}_R}(\pi^\Psi) + \beta * (C_{\mathcal{T}(\mathcal{M}_H, \mathcal{E})}^\Psi(\hat{\pi}) - C_{\mathcal{T}(\mathcal{M}_H, \mathcal{E})}^\Psi(\pi_2^*)))$
   *where $\pi_1^* \in \Pi_{\mathcal{T}(\mathcal{M}_H, \hat{\mathcal{E}})}^*$ and $\pi_1^* \in \Pi_{\mathcal{T}(\mathcal{M}_H, \mathcal{E}^\Psi)}^*$ and $\beta$ captures the degree of penalty imposed by inexplicability of the given plan.*

While the current work uses the difference of cost as the measure of inexplicability, one could have use other plan distances (or a combination) as standin for this measure. This is especially helpful when the user's model/preferences are not completely known.

# 5 Additional Considerations

Now that we have defined some of the problems and solution concepts that have been covered in earlier works, we will now cover some additional problem considerations related to multi-model planning that has been previously ignored.

## 5.1 The Stage of Interaction

An important factor we have more or less ignored in previous sections (and for that matter most of the previous works in model reconciliation) is whether the system is providing an explanation for a plan that has it has proposed or is the system providing post-hoc explanation for some plan that it has just executed (we will refer to such scenarios as explaining behaviors). One may be tempted to argue that this distinction is unnecessary if we are limiting our attention to completely specified sequential plans in deterministic domains, but the fact that the user is viewing a plan being executed gives us opportunities to simplify explanation that may just not be available when the user and the system are deliberating over a proposed plan. In particular, we could use the agents actions (the fact they were successfully executed and their effects) to help shape the user's perception about the task and the agent's capabilities. For example, the robot opening a door is enough to inform the human that the door was not

locked in the first place and does not require a separate communication action or the robot could go through a passage to show to the human that the passage was not blocked. Thus in these multi-model settings, we need to allow for the fact that these actions not only have an effect on the task level state (i.e ontic effects) but may also have epistemic effects on the user's mental models. We could go one further step and treat all explanations as executing actions with epistemic effects. This means that model-reconciliation planning is in fact an epistemic planning problem. Previous works such as [Muise et al., 2015] have shown how certain subsets of epistemic planning problem can be compiled to classical planning problems. In the following sections, we will discuss how we can leverage similar compilation schemes to capture multi-model planning problems and show how such methods provide us with computational advantages over methods that separate reasoning about explanations and the task level plans.

## 5.2 User's Attention

An assumption made by many of the earlier works is the fact that the human observer is a perfect listener. Which means that once the explanation is provided she will definitely include it in her reasoning. Unfortunately, this is not true in most cases. The human's ability to understand the explanation may depend on factors like the hardness of the concept being explained (for eg: explaining the robot's reach in terms of motion constraints vs the ability for the robot to pick up heavy objects) and the mode of explanation (a simple visualization vs natural language) and even the cognitive load of the listener. A more realistic approach would require us to model the fact that the user may not necessarily consider some information even when it is presented to them. The system may need to repeat its explanatory messages or use simpler explanations. Computing such strategies would require us to move to formulations that allows for non-deterministic or stochastic transitions for explanatory messages. In subsection 6.4, we discuss how our new formulation can be extended to support such scenarios.

# 6 Compilation

To support planning with explanatory actions, we will adopt a formulation that is similar to the one introduced in [Muise et al., 2015] to compile reasoning about epistemic states into a classical planning problem.In our setting, each explanatory actions can be viewed as an action with an epistemic effect. One interesting distinction to make here is the fact that the human's belief state now not only includes their belief about the task state but also their belief about the robot's model. This means that the planning model will need to separately keep track of (1) the current robot state, (2) the human's belief regarding the current state, (3) how actions would effect each of these (as humans may have differing expectations about the effects of each action) and (4) how those expectations change with explanations.

Given the model reconciliation planning problem $\Psi = \langle \mathcal{M}_R, \mathcal{M}_H \rangle$, we will generate a new planning model $\mathcal{M}_\Psi = \langle F_\Psi, A_\Psi, I_\Psi, G_\Psi \rangle$ as follows $F_\Psi = F \cup F_\mathcal{B} \cup F_\mu \cup \{\mathcal{G}, \mathcal{I}\}$, where $F_\mathcal{B}$ is a set of new fluents that will be used to capture

the human's belief about the task state and $F_\mu$ is a set of meta fluents that we will use to capture the effects of explanatory actions and $\mathcal{G}$ and $\mathcal{I}$ are special goal and initial state propositions. We will use the notation $\mathcal{B}(p)$ to capture the human's belief about the fluent $p$. We are able to use a single fluent to capture the human belief as we are specifically dealing with a scenario where the human's belief about the robot model is fully known. In this case, we also do not require any of the additional rules that were employed in [Muise et al., 2015] to ensure that the state captures the deductive closure of the agent beliefs.

$F_\mu$ will contain an element for every part of the human model that can be changed by the robot through explanations. A meta fluent corresponding to a literal $\phi$ from the precondition of an action $a$ takes the form of $\mu^+(\phi^{\mathrm{prec}^a})$, where the superscript $+$ refers to the fact that the clause $\phi$ is part the precondition of the action $a$ in the robot model (for cases where the fluent represents an incorrect human belief we will be using the superscript $-$).

For every action $a = \langle \mathrm{prec}^a, \mathrm{adds}^a, \mathrm{dels}^a \rangle \in A_R$ and its human counterpart $a_h = \mathrm{prec}^{a_h}, \mathrm{adds}^{a_h}, \mathrm{dels}^{a_h} \in A_H$, we define a new action $a_\Psi = \langle \mathrm{prec}^{a_\Psi}, \mathrm{adds}^{a_\Psi}, \mathrm{dels}^{a_\Psi} \rangle \in \mathcal{M}_\Psi$ whose precondition is given as –

$$\mathrm{prec}^{a_\Psi} = \mathrm{prec}^{a_R} \cup \{\mu^+(\phi^{\mathrm{prec}^a}) \to \mathcal{B}(\phi) | \phi \in \mathrm{prec}^{a_R} \backslash \mathrm{prec}^{a_H}\}$$
$$\cup \{\mu^-(\phi^{\mathrm{prec}^a}) \to \mathcal{B}(\phi) | \phi \in \mathrm{prec}^{a_H} \backslash \mathrm{prec}^{a_R}\}$$
$$\cup \{\mathcal{B}(\phi) | \phi \in \mathrm{prec}^{a_H} \cap \mathrm{prec}^{a_R}\}$$

The important point to note here is that at any given state, an action in the augmented model is only applicable if the action is executable in robot model and the human believes the action to be executable. Unlike the executability of the action in the robot model (captured through unconditional preconditions) the human's beliefs about the action executability can be manipulated by turning the meta fluents on and off.

The effects of these actions can also be defined similarly. In addition to these task level actions (represented by the set $A_\tau$), we can also define explanatory actions ($A_\mu$) that either add $\mu^+(*)$ fluents or delete $\mu^-(*)$.

Special actions $a_0$ and $a_\infty$ that are responsible for setting all the initial state conditions true and checking the goal conditions are also added into the domain model. $a_0$ has a single precondition that checks for $\mathcal{I}$ and has the following add and delete effects –

$$\mathrm{adds}^{a_0} = \{\top \to p \mid p \in I_R\} \cup \{\top \to \mathcal{B}(p) \mid p \in I_H\}$$
$$\cup \{\top \to p \mid p \in F_{\mu^-}\}$$

$$\mathrm{dels}^{a_0} = \{\mathcal{I}\}$$

where $F_{\mu^-}$ is the subset of $F_\mu$ that consists of all the fluents of the form $\mu^-(*)$. Similarly, the precondition of action $a_\infty$ is set using the original goal and adds the proposition $\mathcal{G}$.

$$\mathrm{prec}^{a_\infty} = G_R \cup \{\mu^+(p^G) \to \mathcal{B}(p) \mid p \in G_R \backslash G_H\} \cup$$
$$\{\mu^-(p^G) \to \mathcal{B}(p) \mid p \in G_H \backslash G_R\} \cup \{\mathcal{B}(p) \mid G_H \cap G_R\}$$

Finally the new initial state and the goal specification becomes $I_\mathcal{E} = \{\mathcal{I}\}$ and $G_\mathcal{E} = \{\mathcal{G}\}$ respectively. To see how such a compilation would look in practice, consider an action (move_from p1 p2) that allows the robot to move from point p1 to p2 only if the path is clear. The action is defined as follows in the robot model.

```
(:action move_from_p1_p2
    :precondition (and (at_p1)
                       (clear_p1_p2))
    :effect (and (not (at_p1))
                 (at_p2) ))
```

Let's assume the human is aware of this action but doesn't know that they need to care about the status of the path (as they assume the robot can move through any debris filled path). In this case, the corresponding action in the augmented model and the relevant explanatory action will be

```
(:action move_from_p1_p2
    :precondition
    (and
        (at_p1) (B((at_p1))) (clear_p1_p2)
        (implies
            (μ+_prec(move_from_p1_p2,
                (clear_p1_p2)))
            (B((clear_p1_p2))) ))
    :effect
    (and (not (at_p1)) (at_p2)
         (not B(at_p1))
         B(at_p2)) ))

(:action explain_μ+_prec_move_from_clear
    :precondition
    (and )
    :effect
    (and μ+_prec(move_from_p1_p2,
        (clear_p1_p2)) ))
```

We will refer to an augmented model that contains an explanatory action for each possible model update and has no actions with effects on both the human's mental model and the task level state as the **canonical augmented model**.

Given an augmented model, let $\pi_\mathcal{E}$ be some plan that is valid for this model ($\pi_\mathcal{E}(I_\Psi) \subseteq G_\Psi$). From $\pi_\mathcal{E}$, we extract two types of information – the model updates induced by the actions in the plan (represented as $\mathcal{E}(\pi_\mathcal{E})$) and the sequence of actions that have some effect of the task state (henceforth referred to as actions with ontic effects) represented as $\mathcal{D}(\pi_\mathcal{E})$. Note that $\mathcal{E}(\pi_\mathcal{E})$ may contain effects from action in $\mathcal{D}(\pi_\mathcal{E})$. This brings us to the following proposition –

**Proposition 1.** *Given a multi-model planning problem $\Psi$ and the corresponding augmented model $\mathcal{M}_\Psi$, then for any plan $\pi(I_\Psi) \models_\Psi G_\Psi$, the tuple $\langle \mathcal{E}(\pi), \mathcal{D}(\pi) \rangle$ is a valid solution for $\Psi$.*

This result can be trivially shown to be true given the above formulation. Unfortunately, the compilation on it's own only takes care of generating plans along with the justifications for correctness of the plan. In many cases, the user would also be interested in understanding why the given plan is optimal.

**Claim 1.** *Given a multi-model planning problem $\Psi$ and the corresponding augmented model $\mathcal{M}_\Psi$, then there exists a plan $\pi(I_\Psi) \models_\Psi G_\Psi$, such that the tuple $\langle \mathcal{E}(\pi), \mathcal{D}(\pi) \rangle$ is a complete solution for $\Psi$ but $\pi$ may not be optimal for $\mathcal{M}_\Psi$.*

The first part of the claim comes from the fact that the space of valid plans for $\mathcal{M}_\Psi$ spans the entire set of valid plans for the robot model and the set of all possible model updates. On the other hand, due to the structure of the preconditions, for a given a set of model updates, there may be plans that are valid in the human model that never gets expanded. This means when the search comes up with a plan $\pi^*$ that is optimal for $\mathcal{M}_\Psi$, it is possible that $\mathcal{M}_H + \mathcal{E}(\pi^*)$ could have plans cheaper than $\mathcal{D}(\pi^*)$, that were not expanded as they were invalid in the robot model.

## 6.1 Planning For Complete Solutions Using Augmented Model

A way to generate complete solutions would be by updating the goal test used by the search. In addition to checking if the goal facts are indeed met in the resultant state, we would now also need to check that the plan is in fact optimal in the updated model. Note that this is an inversion of the search in [Chakraborti et al., 2017] with the added advantage that we are explicitly reasoning with explanatory actions and only check for plan optimality in human models that are guaranteed to support at least one robot executable plan. Furthermore, if we memoize the results of each secondary search with respect to $\mathcal{E}(\pi)$, we can guarantee that the number of optimality tests will be less than or equal to the number of tests required by the earlier approach. Given the nature of this suboptimality test, it should be possible to leverage methods like search space reuse to speed up search, but since our focus here is on establishing the properties of the simplest formulation we will focus on cases that use a simple optimality test.

The next question to ask would be, under what conditions can this new encoding be guaranteed to generate explanations that are minimally complete. Before we formally state the conditions, let us define a new concept called **optimality gap** (denoted as $\Delta\pi_\mathcal{M}$) for a planning model, which captures the cost difference between the optimal plan and the second most optimal plan. $\Delta\pi_\mathcal{M}$ can be defined as –

$$\Delta\pi_\mathcal{M} = \max\{v \mid v \in \mathbb{R} \wedge \nexists\pi_1, \pi_2((0 < (C(\pi_1) - C(\pi_2)) < v) \wedge \pi_1(I_\mathcal{M}) \models_\mathcal{M} G_\mathcal{M} \wedge \pi_2(I_\mathcal{M}) \in \Pi_\mathcal{M}^*\}$$

**Theorem 1.** *Given a canonical augmented model $\mathcal{M}_\Psi$ for a multi-model planning problem $\Psi = \langle \mathcal{M}_R, \mathcal{M}_H, \mathcal{T} \rangle$, if the sum of costs of all explanatory actions is less than or equal to $\Delta\pi_{\mathcal{M}_R}$ and if $\pi$ is the cheapest valid plan for $\mathcal{M}_\Psi$ such that $\mathcal{D}(\pi) \in \Pi^*_{\mathcal{T}(\mathcal{M}_\Psi, \mathcal{D}(\pi))}$, then*

*(1) $\mathcal{D}(\pi)$ is optimal for $\mathcal{M}_R$*

*(2) $\mathcal{E}(\pi)$ is the MCE for $\mathcal{D}(\pi)$*

*(3) There exists no plan $\hat{\pi} \in \Pi_R^*$ such that MCE for $\mathcal{D}(\hat{\pi})$ is cheaper than $\mathcal{E}(\pi)$ ,i.e, the search will find an optimal explicable plan if one exists.*

*Proof Sketch.* We observe that there exists no valid plan $\pi'$ for the augmented model ($\mathcal{M}_\Psi$) with a cost lower than that of $\pi$ and where the ontic fragment ($\mathcal{D}(\pi')$) is optimal for the human model. Let's assume $\mathcal{D}(\pi) \notin \Pi_\mathcal{R}^*$ (i.e current plan's ontic fragment is not optimal in robot model) and let $\hat{\pi} \in \Pi_\mathcal{R}^*$. Now let's consider the augmented plan corresponding to $\hat{\pi}$, $\hat{\pi}_\mathcal{E}$, i.e, $\mathcal{E}(\hat{\pi}_\mathcal{E})$ is the MCE for the plan $\hat{\pi}$) and $\mathcal{D}(\hat{\pi}_\mathcal{E}) = \hat{\pi}$. Then the given augmented plan $\hat{\pi}_\mathcal{E}$ is a valid solution for our augmented planning problem $\mathcal{M}_\Psi$ (since the $\hat{\pi}_\mathcal{E}$ consists of the MCE for $\hat{\pi}$, the plan must be valid and optimal in the human model), moreover the cost of $\hat{\pi}_\mathcal{E}$ must be lower than $\pi$. This contradicts our earlier assumption hence we can show that $\mathcal{D}(\pi)$ is in fact optimal for the robot model.

Using a similar approach we can also show that no cheaper explanation exists for $\pi_\mathcal{E}$ and there exists no other plan with a cheaper explanation. $\square$

Also note, that while it is hard to find the exact value for the optimality gap, we are guaranteed that the optimality gap is greater than or equal to one for domains with only unit cost actions or is guaranteed to be greater than or equal to $(C_2 - C_1)$, where $C_1$ is the cost of the cheapest action and $C_2$ is the cost of the second cheapest action (i.e $\forall a, (C_\mathcal{M}(a) < C_2 \rightarrow C_\mathcal{M}(a) = C_1)$)

## 6.2 Balanced Plans

We can use the above formulation directly to obtain balanced explicable plans, we just no longer need to put any specific restriction on the cost of explanatory action. To generate optimal balanced plans, we need to relax the requirement that the final plan is optimal in the human model. Instead we can incorporate the inexplicability penalty into the reasoning about the plan, by assigning the cost of $a_\infty$ (the goal action) to be $\beta$ times the difference between the optimal plan in the human model and the current plan. When $\beta$ is set to zero the problem would just identify the optimal plan corresponding to the original robot model and when $\beta$ is set high enough the formulation just generates explicable balanced plans. This can be guaranteed if beta is set higher that $\kappa$, where $\kappa$ is some upper bound on plan length for the robot (that includes explanatory actions). On the other hand, if the cost of any explanatory actions is also higher than $\kappa$ then the formulation will try to find the plan that is closest to an optimal plan in the original human model and is still executable in the robot model.

To see the use of balanced plans, lets revisit the urban search and rescue case. Here, the optimal path for the robot to follow would be to go through P7, P8 and P12. The human thinks the path should be the one through P6 and P5. Explaining the optimality of the path requires explaining that the path from P6 to P5 is blocked (which can be explained through the action explain_obstructed_P6_P5) and the path from P8 to P12 is clear (explained by explain_away_obstructed_P8_P12). Let us assume that the application of each of these explanatory actions increases the total cost by ten. The balanced explicable plan in this setting would be

```
explain_obstructed_P6_P5→INIT_ACT→
move_from_p1_p9→clear_passage_p9_p10→
move_from_p9_p10→move_from_p10_p11→
move_from_p11_p16→move_from_p16_p15→
```

GOAL_ACT

If we try to identify an optimal balanced plan for $\beta = 1$ and a cost of 5 for the clear passage action, then the plan that would be generated would be –

INIT_ACT→
move_from_p1_p9→clear_passage_p9_p10→
move_from_p9_p10→move_from_p10_p11→
move_from_p11_p16→move_from_p16_p15→
GOAL_ACT

### 6.3 Plans with Epistemic Side-effects

As mentioned earlier, in cases where the user is observing a plan being executed, even the agent's non-explanatory actions could have effects on user's mental model. We can easily incorporate this requirement by associating effects involving meta-fluents into our task specific actions. Such effects may be specified by domain experts or could be generated using heuristic rules. For example, if an action is executed in a state where a precondition believed by the user is not met then that precondition should be removed from the human's perceived model.

To illustrate the use of such explanatory actions in our encoding let us visit the USAR scenario and assume that the human thinks that the path from P8 to P12 is blocked and the one from P6 to P5 is free. Also in this setting, for explaining the status of passages (whether they are blocked or not) the robot can now use two actions, one a rather expensive explicit communication action, that sends the updated map information to the human or the robot can just visit the blocked passage and the human who is watching a video feed of robot actions will learn that the passage is blocked or clear. Thus the action descriptions for the move action will be –

```
(:action move_from_P7_P8
    :precondition (and (robot_at_P7)
        ...
        (B(robot_at_P7)))
    :effect (and (robot_at_P8)
        ...
        (B(clear_P8_P12))
        (increase (total-cost) 1)))
```

With this new action the robot knows that as soon as it reaches P8 the human would know that the path from P8 to P12 is clear so it can continue on that path. So the new robot plan will be –

INIT_ACT→move_from_p1_p7→move_from_p7_p8→
move_from_p8_p12→move_from_p12_p13→
explain_obstructed_p6_p5→
move_from_p12_p13→move_from_p13_p14→
move_from_p14_p15→GOAL_ACT

### 6.4 Plans for Inattentive Users

In this scenario, we can no longer assume that explanatory actions would have deterministic effects on user's model and that means considering planning models that allow for non-deterministic or stochastic effects.

| Domain | New Compilation | | Model Space Search | |
|---|---|---|---|---|
| | cov. | runtime | cov. | runtime |
| Blocksworld | 13/15 | 569.38 | 13/15 | 2318.73 |
| Elevator | 15/15 | 59.20 | 1/15 | 3382.462 |
| Gripper | 5/15 | 2301.90 | 6/15 | 2093.54 |
| Driverlog | 4/15 | 2740.38 | 2/15 | 3158.59 |
| Satellite | 2/15 | 3186.93 | 0/15 | 3600 |

Figure 3: Table showing average runtime (sec) and coverage for explanations generated for standard IPC domains.

To see a simple example of how this would look, consider the USAR domain and look at the ability of the move action to inform the commander about the status of the passage from P8 to P12. The robot can not always guarantee that the commander would be looking at the screen and there is a chance that the commander won't be looking at the screen when the path is presented. Thus in the new definition of action (move_from_P7_P8) we will replace the effect that adds $\mathcal{B}$(clear_P8_P12) with the effect (one-off ($\mathcal{B}$(clear_P8_P12)) (and)) Which means that the action's ability to update the human model is a non-deterministic effect. In this case, we can look for a conformant plan by converting the earlier search into a search over belief space. A search node only passes the goal test if the goal condition are met in every state in the belief space. Thankfully, this particular problem does have a conformant solution –

explain_obstructed_p6_p5→
explain_clear_p8_p12→INIT_ACT→
move_from_p1_p7→move_from_p7_p8→
move_from_p8_p12→move_from_p12_p13→
move_from_p13_p14→move_from_p14_p15→
GOAL_ACT

## 7 Empirical Comparison of Model-Space Search and Planning with Explanatory Actions

The focus of this section is to see how our compilation compares with the approaches that separate the reasoning about explanations and plan generation. In particular, we will consider the approaches discussed in [Chakraborti, Sreedharan, and Kambhampati, 2018b] as a point of comparison. Note that in order for the model space search to always identify the optimal balanced explicable plan, generating an optimal plan at each possible model is not enough. The approach would require iterating over the space of all possible optimal plans at a given node to find one that is executable in the robot model or require involved compilations that only produce optimal plans that are executable in robot model. To avoid changing the method too much, we used an optimistic version of the model space search that only identifies one optimal plan per search node and the search ends as soon as it find a node where the optimal plan produced has the same cost as the robot's plan and is executable in the robot model.

For comparison, we selected five IPC domains and for each domain, we created three unique models by introducing 10

random updates in the model (except in the case of gripper and driverlog where only 5 were removed). Each of these three domains were paired with five problem instances and then tested on each of the possible configurations. Each instance was run with a limit of 30 minutes, all explanatory actions were restricted to the beginning of the plan and the cost of explanatory actions were set to be twice the cost of original action. Table 3 lists the time taken to solve each of these problems. For calculating the average runtime, we used 1800 secs as the stand in for the runtime of all the instances that timed out. We used h_max as the heuristic for all the configurations.

As clearly apparent from the table, the new approach does better than the original method for generating balanced plans for most of the domains. Gripper seems to be the only domain, where model search seem to be doing better but this is also a domain that had the smallest number of model differences. This points to the fact that the ability to leverage planning heuristics seems to make a marked difference in domains with a large number of possible explanatory actions.

## 8 Related Work

It's widely accepted in social sciences literature that explanations must be generated while keeping in mind the beliefs of the agent receiving the explanation [Miller, 2018; Slugoski et al., 1993]. As such, epistemic planning makes for an excellent framework for studying the problem of generating these explanations. While the most general formulation of epistemic planning has been shown to be undecidable, many simpler fragments have been identified [Bolander, Jensen, and Schwarzentruber, 2015]. Recently, there have been a lot of interest in developing efficient methods for planning in such domains [Muise et al., 2015; Kominis and Geffner, 2015; 2017; Le et al., 2018; Huang et al., 2018]. In our base scenario, we will assume (1) a finite nesting of beliefs, (2) the human is merely an observer, and (3) all actions are public. The specific problems discussed in our paper hardly exercises most of the capabilities provided by epistemic planning. It's important to note that given the epistemic nature of the explanatory actions, solving the general model reconciliation problem would require leveraging all those capabilities. Our hope is that by presenting model reconciliation in this more general setting, the community would be motivated to start looking at more general and complex versions of these problems.

Our work also looks at the use of explanatory actions as a means of communicating information to the human observer. The most obvious types of such explanatory action includes purely communicative actions such as speech [Tellex et al., 2014] or the use of mixed reality projections [Chakraborti, Sreedharan, and Kambhampati, 2018a; Ganesan, 2017], but recent works have shown that physical agents could also use movements to relay information such as intention [MacNally et al., 2018; Dragan, Lee, and Srinivasa, 2013] and incapability [Kwon, Huang, and Dragan, 2018]. Our framework could be easily adopted to any of these explanatory actions and would naturally allow for a trade-off between these different types of communication.

Many recent works dealing with explanation generation for planning, have looked at characterizing explanation in terms of the types of questions they answer (c.f [Fox, Long, and Magazzeni, 2017; Smith, 2012] and contrastive explanations in general). This characterization is orthogonal to the question of what type of information constitutes valid explanations. Putting aside questions regarding observability, the reason why a user requests an explanation is either due to knowledge asymmetry (incomplete or incorrect knowledge of the task) or due to limitations of their inferential capabilities. Depending on the context, the answer to any of the questions described in these papers would require correcting human's model of the task and/or providing inferential assistance. Works that have looked at model reconciliation explanations have mostly focused on the former. Explanations discussed in this paper can be viewed as an answer to the question "Why this plan?" (which can also be viewed as a contrastive question of the form "Why this plan and not any other plan?"). This is not to say that in complex scenarios just the model reconciliation information would suffice but it would need to be supplemented with information that can bridge the differences in inferential capabilities. Use of abstractions [Sreedharan, Srivastava, and Kambhampati, 2018], providing refutation of specific foils [Sreedharan, Srivastava, and Kambhampati, 2018] and providing causal explanations [Seegebarth et al., 2012] could all be used to augment model reconciliation explanations.

## 9 Conclusion and Discussion

The paper presents a more general formulation for the problem of planning with users in the loop with asymmetric models than any of the previous works. We discuss how this formulation can be extended to capture novel explanatory behaviors and can be solved using approaches that are computationally more efficient than methods that rely on direct model space search. One possible avenue for future work would be investigating and implementing planning compilations that capture extensions of the model reconciliation problem that have previously been investigated like specific foils, uncertain human models, state abstractions, differences in action costs, disjunctive preconditions, etc... It would also be worth investigating if there are any specific considerations to be made when choosing heuristics for such planning models.

## Acknowledgments

This research is supported in part by the ONR grants N00014-16-1-2892, N00014-18-1-2442, N00014-18-1-2840, the AFOSR grant FA9550-18-1-0067, NASA grant NNX17AD06G and JP Morgan faculty research grant.

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
