# OpenReview forum: "A General Framework for Synthesizing and Executing Self-Explaining Plans for Human-AI Interaction"
_icaps-conference.org/ICAPS/2019/Workshop/XAIP — XAIP 2019_

### Official Review · AnonReviewer1 · 2019-05-09
**Great paper on model reconciliation with explanation actions**

**Rating:** 4
**Confidence:** 2

**Review:**

The paper is generally well-written and clear. The primary contribution is a formulation of human-in-the-loop planning problems that (1) captures existing approaches and (2) provides a computational advantage over previous approaches.

Deciding when to perform explanation as part of the planning process (during plan generation or execution) is a novel consideration.

The paper could expand on how the user's model could be extracted since it is a core requirement of this approach.

This paper is highly relevant to the workshop and should lead to fruitful discussion.

Small comments/grammar:

7: "many simpler fragments has been.." --> "many simpler fragments have been...."
7: "is either due knowledge asymmetry..." --> "is either due to knowledge asymmetry..."

---

### Official Review · AnonReviewer3 · 2019-05-13
**great and highly relevant work; bit overloaded**

**Rating:** 4
**Confidence:** 2

**Review:**

The paper discusses variants of multi-model planning problems, one model being the actual model and the other being the human's model, where a plan also requires explanations -- model-changing actions -- to clarify to the human any model differences required to (be able to) understand that the returned plan is optimal. A major novelty is the realization that executing actions may in itself affect the human's model, so that the overall planning process has an epistemic nature. A combined (compiled) mode is proposed to this end, and various versions of objectives and methods are discussed.

This is definitely a great paper for XAIP and should be accepted. I personally found the number of notions, notations, and objectives discussed a bit overwhelming. Also the paper is not very self-contained, and basically requires deep and broad nowledge of the author' own prior work. At times, I felt like the authors are talking to themselves rather to an outside reader (e.g.: start of Section 5.1). It would be great if the authors could try to maker theior work more accessible. For presentation in the workshop, I would highly recommend to focus on a few key points rather than trying to bring the entire breadth of this paper across.

I would also note, though, one aspect that seems to be missing from the discussion: the "explanations" considered here are a long way away from what could be considered a complete explanation of the question "why this plan and not any other plan?". The problem of inferential capability is a huge one (how to summarize to a human the main reasons for a decision taken based on complex reasoning processes), and in addition to this there is the issue of why the system came up with this specific solution among many possible cost-equivalent ones. In my mind, these things are actually (among) the major issues addressed in XAI. I would prefer if these issues could be emphasized a bit more, than be mentioned almost as an afterthought at the end of the related work discussion.

I think the introduction should already cite the works it refers to.

Write-up is good in general. but the authors could try to simplify their notations somewhat, and/or take care to accompany each notation with a brief natural language description. Understanding Definition4, for example, is painful as is. I would also recommend to move Section 4 further up front, it's awkward to read in Section 3.1 about whats going to happen next and then have to read 3.2 and 4 until it actually happens.

---

### Decision · Program_Chairs · 2019-05-15

**Decision:**

Accept

**Comment:**

The reviewers agree to accept. Please address all review criticism as best possible for the final paper version and its presentation at the workshop. Looking forward to discuss your work at the workshop!